# Lipid Membrane State Change by Catalytic Protonation and the Implications for Synaptic Transmission

**DOI:** 10.3390/membranes12010005

**Published:** 2021-12-21

**Authors:** Christian Fillafer, Yana S. Koll, Matthias F. Schneider

**Affiliations:** Medical and Biological Physics, Department of Physics, Technical University Dortmund, Otto-Hahn-Str. 4, 44227 Dortmund, Germany; yana.koll@tu-dortmund.de (Y.S.K.); matthias-f.schneider@tu-dortmund.de (M.F.S.)

**Keywords:** acetylcholine, acetic acid, proton, pH, synapse, acetylcholine receptor (AChR), cholinergic, postsynaptic excitation

## Abstract

In cholinergic synapses, the neurotransmitter acetylcholine (ACh) is rapidly hydrolyzed by esterases to choline and acetic acid (AH). It is believed that this reaction serves the purpose of deactivating ACh once it has exerted its effect on a receptor protein (AChR). The protons liberated in this reaction, however, may by themselves excite the postsynaptic membrane. Herein, we investigated the response of cell membrane models made from phosphatidylcholine (PC), phosphatidylserine (PS) and phosphatidic acid (PA) to ACh in the presence and absence of acetylcholinesterase (AChE). Without a catalyst, there were no significant effects of ACh on the membrane state (lateral pressure change ≤0.5 mN/m). In contrast, strong responses were observed in membranes made from PS and PA when ACh was applied in presence of AChE (>5 mN/m). Control experiments demonstrated that this effect was due to the protonation of lipid headgroups, which is maximal at the pK (for PS: ${\mathrm{pK}}_{\mathrm{COOH}}\approx 5.0$; for PA: ${\mathrm{pK}}_{{\mathrm{HPO}}_{4}^{-}}\approx 8.5$). These findings are physiologically relevant, because both of these lipids are present in postsynaptic membranes. Furthermore, we discussed evidence which suggests that AChR assembles a lipid-protein interface that is proton-sensitive in the vicinity of pH 7.5. Such a membrane could be excited by hydrolysis of micromolar amounts of ACh. Based on these results, we proposed that cholinergic transmission is due to postsynaptic membrane protonation. Our model will be falsified if cholinergic membranes do not respond to acidification.

## 1. Introduction

In biological systems, different mechanisms have evolved so that stimuli from the environment (chemical, mechanical, optical, etc.) can be received. In multicellular organisms, this includes the means by which neighbouring cells can sense each other. In a chemical synapse, for example, one cell releases a substance which subsequently affects its neighbour(s). One of the most prevalent classes of synapses uses acetylcholine (ACh) as the neurotransmitter. Upon its release, ACh is rapidly hydrolyzed by synaptic enzymes (acetylcholinesterase (AChE), butyrylcholinesterase (BuChE) and other non-specific esterases) [1]. The turnover number of AChE is among the highest of all enzymes found in nature (∼25,000 s${}^{-1}$). From early on, it has been believed that this catalytic activity serves the purpose of destroying the neurotransmitter after it had bound to a receptor protein [2]. Since the 1970s, evidence for such a receptor protein has accumulated. A main line of argument has involved the experimental use of non-hydrolyzable cholinergic agonists and toxins such as α-neurotoxins. The latter is a class of polypeptides, which inhibit miniature synaptic potentials [3], neuromuscular transmission [4], and agonist-induced ion-flux response in postsynaptic membrane vesicles [5]. The inhibitory action is realized at concentrations of toxin lower than those that are required to affect catalysis by AChE [5]. When homogenates of electrocytes were chromatographically separated on α-neurotoxin-functionalized affinity resin, a protein now known as AChR was obtained [6,7,8,9]. Importantly, these purification studies also demonstrated that AChR is not identical with AChE [6]. It was further shown that antibodies against AChR impair neuromuscular transmission [7]. Finally, when the genes coding for the subunits of AChR were expressed in oocytes, these cells became sensitive to ACh [10]. These pieces of evidence indicate that AChR can optimize the sensitivity of a cell membrane for ACh. Nevertheless, the proposed mechanism - ligand-gated opening of a proteinous pore - has remained a hypothesis.

In the 1980s, K. Kaufmann suggested in a series of works that postsynaptic excitation relies on protons, which are liberated during the hydrolysis of ACh [11,12,13,14]. It was demonstrated that protons can indeed induce transmembrane currents in lipid bilayers and that the channel opening probability is maximal at the protonation transition of the lipid headgroups [15]. Motivated by Kaufmann’s works, we recently found that excitable cells can become responsive to ACh in the absence of AChR [16]. When ACh was catalytically hydrolyzed, the cell membrane depolarized and action potentials were triggered. It was demonstrated that the cells are excited by protons which dissociate from the hydrolysis product acetic acid (AH). Taken together, these findings are intriguing, in particular in light of the fact that many neurons have also been shown to be sensitive to acids (e.g., [17,18,19]).

Herein, the response of lipid membrane interfaces to ACh was investigated further. The central role of the catalyst in generating a strong membrane state change was confirmed. The composition of postsynaptic membranes was reviewed and molecules with physiologically relevant pH sensitivity were identified. It was proposed that AChR is involved in membrane excitability by a different mechanism, namely by setting up a proton-sensitive membrane interface. Finally, fundamental properties of the postsynaptic membrane (pH dependency, number of channel openings, excitatory current time scales) were interpreted in light of the membrane protonation concept.

## 2. Materials and Methods

### 2.1. Materials

1,2-dimyristoyl-sn-glycero-3-phospho-L-serine (DMPS), 1,2-dimyristoyl-sn-glycero-3-phosphatidic acid (DMPA), 1,2-dipalmitoyl-sn-glycero-3-phosphatidylcholine (DPPC) were obtained from Avanti Polar Lipids. Acetylcholinesterase from *E. electricus* (Type V-S and Type VI-S, specific activity ≥1 Unit/μg) was purchased from Sigma Aldrich. All other reagents were obtained from Sigma-Aldrich and were of analytical purity (≥99%).

### 2.2. State Diagram of Membrane Interface at Constant Area

A crystallizing dish with an inner cross-sectional area of ≈34 cm${}^{2}$ was thoroughly cleaned with water and isopropanol. 70 mL of standard subphase (100 mM NaCl with 0.5 mM NaH${}_{2}$PO${}_{4}$ for pH 7.0 or with 0.5 mM TRIS for pH 10.0) were added and the solution was stirred at <100 rpm. A Wilhelmy plate mounted on a surface pressure sensor (PS4; NIMA) was used to continuously monitor the surface tension (γ) of the air-solution interface. The surface pressure (π) as reported herein is defined as the difference between the surface tension without and with added lipid $\left(\pi =\gamma_{0}-\gamma \right)$.

The pH of the subphase was monitored with a pH sensor (LE422; Mettler Toledo) and the temperature (T) of the setup was regulated to 23.0 °C (DMPS) or 33.0 °C (DMPA). After equilibration for 30 min, a solution of DMPS or DMPA in chloroform:methanol:water (65:35:7; *c* = 1 mg/mL) was added in 0.4–1 μL increments with a Hamilton syringe fitted with a repeating dispenser (PB600−1). Following every addition of lipid, the system was equilibrated for 2–15 min.

### 2.3. Protonation Transition of Membrane Interface

DMPS was titrated onto standard subphase with pH 7 at T≈ 23.0 °C until the transition pressure was reached ($\pi_{T}\approx 22$ mN/m). After equilibration for 30–60 min, the pH of the subphase was lowered in 0.5 unit increments by addition of HCl (c=1 M) or CH${}_{3}$COOH (c=1 M from pH 7 to 4.5 and c=17.4 M from pH 4.5 to 3.0). In the case of DMPA, the lipid was titrated onto standard subphase with pH 10 at T≈ 33.0 °C until $\pi_{T}\approx 29.0$ mN/m was reached. Following equilibration, the monolayer was titrated in a stepwise manner with 1 M HCl from pH 10.0 to 7.5. After every titration step, the system was equilibrated for 5–15 min. In vicinity of the pK, the lateral pressure did not equilibrate fully. This was probably due to the detachment of lipid into the subphase.

### 2.4. Membrane Response upon Injection of Acetylcholine (ACh)

DMPS was titrated onto a standard subphase until a state in the liquid-expanded phase close to the main transition was reached (π≈20 mN/m; see red circle in Figure 1). After equilibration for 60 min, 14 μL of a stock solution of AChE ($c=5\;\frac{\mathrm{Units}}{\mu L}$; final concentration in subphase: c≈5 nM), equimolar stock solution of bovine serum albumin (BSA, non-catalyzed control) or subphase (non-catalyzed control) was added. Following equilibration for an additional 10 min, 160 μL of a solution of ACh in subphase (c=4.44 M) were injected from ∼1 mm above the air-solution interface (final concentration in subphase: c=10 mM). To study the effect of the hydrolysis products on the membrane interface, a monolayer was prepared as described above without AChE. 160 μL of a 4.44 M solution of acetic acid or choline were injected (final concentration in subphase: c=10 mM). Since injection of any fluid volume >50 μL led to transient surface pressure changes, separate control experiments were conducted. In these experiments, a monolayer without AChE was prepared and 160 μL of subphase were injected. The changes of π associated with this “injection artifact” were averaged over n=3 experiments (Appendix A) and were subtracted from all other injection experiments.

For the injection experiments with DMPA, lipid was titrated until a lateral pressure of ≈24 mN/m was reached at pH 9 and T≈ 33.0 °C. Following equilibration and addition of AChE, 160 μL of a solution of ACh were injected (final concentration in subphase: c=100
μM).

For the injection experiments with DPPC, lipid was titrated until a lateral pressure of ≈16 mN/m was reached at pH 7 and T≈30.0 °C (the main transition pressure under these conditions was $\pi_{T}\approx 18$ mN/m). Following equilibration and addition of AChE, 160 μL of a solution of ACh were injected (final concentration in subphase: c=10 mM). In all of the injection experiments the data recording frequencies were: surface pressure (100 Hz), pH (10 Hz), temperature (10 Hz).

### 2.5. Adsorption of AChE to the Gas-Solution Interface

The standard subphase was equilibrated for 30 min at T≈23.0 °C. Subsequently, 7 μL of either a solution of AChE ($c=5\;\frac{\mathrm{Units}}{\mu L}$) or subphase (control) were injected. For the experiment at pH 4, the subphase had been acidified with hydrochloric acid.

## 3. Results and Discussion

### 3.1. Catalytic Hydrolysis of ACh Triggers Membrane Response

A simple model of a cell membrane was prepared by titrating the phospholipid DMPS onto an air-water interface at constant area. State diagrams of the membrane interface were readily obtained (Figure 1). Based on the diagrams, the pressure range of the main transition ($\pi_{T}$) can be identified and liquid-expanded and liquid-condensed states can be delineated (“fluid” and “gel” respectively in bilayer terminology). Under the conditions employed, $\pi_{T}\approx 22$ mN/m which is in close agreement with studies of DMPS films on Langmuir-Blodgett troughs [20].

In order to investigate the effect of pulse-like applications of ACh, a membrane state in the liquid-expanded phase close to the main transition was prepared (red circle in Figure 1). This is representative of the resting state of many cell membranes, which also exist on the fluid side of a transition [21]. When ACh was injected (final concentration in subphase: c=10 mM), only negligible changes of π occurred (green diamonds in Figure 2).

This indicated that ACh does not significantly change the state of the PS membrane interface with 100 mM NaCl as background electrolyte. Such a lack of action of ACh on anionic monolayers has been reported previously [22].

An entirely different membrane response was observed when the subphase contained AChE. In the presence of the enzyme, an injection of ACh resulted in a drop of π by several mN/m (Figure 2). In parallel with this initial decrease of pressure, the aqueous subphase acidified. The latter was evidence for the hydrolysis of ACh by AChE, which liberates acetic acid (AH) as well as choline (Ch)
(1)$$ACh^{+}+H_{2}O\;⇌\;AH+Ch^{+}\;$$
and subsequently AH donates a proton to water
(2)$$AH+H_{2}O\;⇌\;A^{-}+H_{3}O^{+}\;$$Over the time course of the experiment, the pH approached an equilibrium value (≈4.4). The observed pH change (ΔpH≈−2.6) was orders of magnitude larger as compared to the non-catalyzed reaction mixture (ΔpH≈−0.1). This emphasizes that the kinetic inhibition of the reaction was lifted by the enzyme. In contrast to pH, the lateral pressure did not equilibrate. ∼500 s after injection, π reached a minimum but then it started to rise for several 1000 s and eventually exceeded its initial level. We suspected that this delayed increase of lateral pressure is due to adsorption of the enzyme to the membrane under acidic conditions. In order to test for this possibility, the interfacial activity of AChE was studied at pH 7 and pH 4 (Appendix A). Over a time course of 100 min at pH 7, adsorption of enzyme resulted in a surface pressure change by 0.6±0.3 mN/m. At pH 4, however, Δπ was almost an order of magnitude larger (3.9±1.4 mN/m). This indicated that the interfacial activity of AChE is indeed higher under acidic conditions. Dziri et al. also reported that AChE assumes a larger area per molecule and becomes surface active in acidic media [23]. Interestingly, the catalytic activity of AChE is reduced under acidic conditions and trends to zero at pH 4 [24]. The assumption of a larger area per molecule and reduction of catalytic activity indicates that the protein undergoes a transition at low pH. This could explain two phenomena in the present experiments (Figure 2): first, the delayed increase of π is probably due to adsorption of denatured AChE. Second, the rather slow equilibration of pH is probably due to a loss of catalytic activity between pH 4 and 5. These conclusions were corroborated by control experiments in which AChE was replaced by BSA. BSA is not enzymatically active towards ACh. As expected, there was neither a significant change of pH nor a significant change of surface pressure of the PS monolayer (Appendix A). Therefore, the biphasic surface pressure response (Figure 2) is not caused by an instability of the monolayer upon the injection of protein.

The most remarkable result of these pulse experiments was the appearance of a membrane response to ACh when AChE was included in the system. An essentially identical result was obtained in experiments with excitable plant cells [16]. There, extracellular application of ACh in absence of AChE did not result in a membrane response. When ACh was applied in presence of AChE, however, the cell depolarized and action potentials were triggered.

### 3.2. Membrane Response Is Due to Lipid Headgroup Protonation

In order to identify the hydrolysis product that led to the membrane response, control experiments were conducted with acetic acid and choline. It was not the goal to track the exact membrane response in Figure 2 as a function of time. Rather, we were concerned with identifying the magnitude and direction of the state change. For this purpose it was sufficient to apply each product at its concentration in the equilibrium mixture. The equilibrium constant for the hydrolysis of ACh is $K=\frac{\left[AH][Ch\right]}{\left[ACh\right]}\approx 140$ [25], which means that products dominate. Therefore, ${\left[Ch\right]}_{eq}={\left[AH\right]}_{eq}\approx {\left[ACh\right]}_{initial}$, i.e., for the present experiment (Figure 2) the final concentration of each hydrolysis product was ≈10 mM. Upon injection of choline at this concentration, the pH of the subphase remained constant and π increased in a step-like manner by ≈+0.5 mN/m (Figure 3a). Therefore, choline can not be responsible for the strong decrease of π that was observed during hydrolysis of ACh. In contrast, injection of acetic acid led to a steep decrease of surface pressure (Δπ≈−12 mN/m) in parallel with acidification of the subphase (ΔpH≈−3.4). There were no indications for a second phase of rising surface pressure in these control experiments. This corroborated that the delayed increase of π (*c.f.* Figure 2) was due to adsorption of denatured AChE. In order to further understand the membrane response to acetic acid, titration experiments were conducted. A DMPS monolayer was prepared in the regime of the main transition. Following equilibration, the subphase was acidified with acetic acid. In the pH range between 3 and 7, pronounced changes of π were observed (Figure 3b). The surface pressure decreased from ≈22 mN/m at pH 7 to ≈8 mN/m at pH 3 in a sigmoidal manner. Based on a host of literature studies [20,26,27,28], this nonlinear decrease of π with acidification can be attributed to protonation of the carboxylic acid moiety of the lipid headgroup. From a sigmoidal fit of the titration curve, the apparent ionization constant of the interface can be extracted. For titration with acetic acid $pK_{COOH}\approx 5.1$. This finding is in close agreement with the literature [20,26,27,28]. In order to investigate if there are any acid-specific effects on the interfacial pK, the titration experiments were repeated with hydrochloric acid. Within experimental error, the progression of the titration curve, inflection point ($pK_{COOH}\approx 4.9$), and final surface pressure were identical (Figure 3b). Therefore, the presence of acetate and protonated acetic acid does not significantly change the π−pH diagram of PS between pH 3 and 7. Based on these experiments, it can be concluded that the decrease of surface pressure during hydrolysis of ACh is chiefly due to lipid protonation. When a membrane is located slightly above or at the pK, it can be considered a “proton receptor” (i.e., the system has a maximal, typcially sigmoidal response to protons) [14].

These control experiments also explained the role of esterase. In the absence of enzyme, ACh hydrolyzed so slowly that the bulk pH remained widely constant. In this case, no membrane state change was effected. When AChE was present, the rate of hydrolysis was increased dramatically. The catalytic liberation of acetic acid generated a pH pulse in the system. The ensuing membrane response was particularly strong when the interface was taken into/across its pK.

#### Implications for the Cholinergic Synapse

Hydrolysis of ACh inevitably occurs in all cholinergic synapses. As demonstrated above, this catalytic reaction can not be called a “deactivation” of the neurotransmitter, because it generates acetic acid. The latter can induce strong state changes in charged membranes. Intriguingly, many neurons have been shown to be pH-sensitive [17,18,19]. Therefore, the possibilility exists that catalytically-liberated protons excite postsynaptic cholinergic membranes. Two open questions are (i) the pH sensitivity of cholinergic membranes and (ii) the magnitude of the acidification pulse during hydrolysis of ACh in a synapse.

### 3.3. pH Sensitivity of the Postsynaptic Membrane

In the following, we will analyze the pKs of postsynaptic membrane molecules. We will mainly focus on the material composition of postsynaptic membranes from *Torpedo* electrocytes, because this is one of the best studied systems. Several independently reported material compositions are available. These have been summarized in Table 1. AChR-rich postsynaptic membranes mainly consist of cholesterol (CHOL), phosphatidylcholine (PC), ethanolamine phosphoglycerides (EPG), and PS on a mol% basis. Minor phospholipid components are sphingomyelin (SM), phosphatidylinositol (PI), cardiolipin (CL) and phosphatidic acid (PA). The protein:lipid ratio is ∼2:1 on a weight basis [8,29,30]. In terms of numbers of molecules, however, lipids clearly outnumber the membrane proteins ∼130:1 and therefore deserve a detailed analysis.

#### 3.3.1. Protonation Transitions of Lipids

The ionization constants of almost all of the lipids in the postsynaptic membrane have been determined in physicochemical studies with mono- and bilayers in salt solutions (Table 1). The following discussion may have to be adjusted for such biological membranes that exist at lower ionic strength or with a transmembrane potential that additionally attracts/repels protons to the interface [15]. When considering the apparent pK values of postsynaptic membrane molecules, it emerges as striking that the main lipids (CHOL, PC, and PE) do not undergo any significant ionization changes in the physiological pH range. CHOL has a non-phenolic hydroxy group for which a very low acidity (pK>10) is expected. The headgroups of PC and PE exist as zwitterions at pH 7–8. The phosphate moieties of these lipids will not be protonated significantly unless the pH in the surrounding medium drops to <5. The same applies to the phosphates of CL, (probably) SM, PI and PS. To be more precise, protonation of phosphates can occur in course of a transient proton pulse at pH 7, but due to the low pKs between 1–3, the groups will only remain protonated for a very short time (proton dwell times are in the 1–100 ns range [42]). In order to exemplify the effect of a low pK on the proton response, we carried out additional control experiments with lipid monolayers made from PC. According to the literature (Table 1), the phosphate of the headgroup of PC has a pK of ∼1. Therefore, a proton pulse generated by hydrolysis of 10 mM ACh, which results in a drop of pH from ∼7 to ∼3–4, was expected to lead to minor protonation only, and therefore only to a minor lateral pressure response. In line with this expectation, an injection of 10 mM acetic acid or of 10 mM ACh in presence of AChE led to a small transient decrease of lateral pressure by ∼1 mN/m (Appendix A). When AChE was present, a delayed increase of lateral pressure by ∼5 mN/m was observed. As in the case of PS (Figure 2), this is due to adsorption of AChE to the interface under acidic conditions. These additional control experiments corroborate our conclusion that the headgroup pK determines the lipid membrane sensitivity to protons and to ACh.

In postsynaptic membranes, there are three phospholipid heads that have protonation transitions that are closer to pH 7. The apparent pK of the carboxy group of PS is ≈4.5–5.4 in pure PS membranes and ≈4 in PC/PE host bilayers [28]. The down-shift of the pK in the mixed membranes is due to the lower surface potential as compared to the pure PS membrane (i.e., −COOH is a better proton donor when the surface potential is low). Two phospholipids which could contribute to a high protonation sensitivity at the physiological pH are PI and PA. For the former, this requires that the inositol is phosphorylated (Table 1). There exists little knowledge, however, regarding the levels of polyphosphoinositides (PIP) in *Torpedo* membranes. Rotstein et al. reported that there are only traces of PIPs in AChR-rich membrane fragments (∼0.1 mol% of phospholipids). The authors noted that the actual quantities could be higher, because loss of these lipids may have occurred during the extraction procedure and because the phosphate esters on the inositol are prone to hydrolysis [31].

PA is peculiar among phospholipids in that it can carry two negative charges on a minimally sized headgroup. The first proton on its phosphate dissociates readily ($pK_{H_{2}PO_{4}}\approx$ 3–4), but the second one has a significantly higher pK. $pK_{HPO_{4}^{-}}$ is ≈8.5 in pure PA membranes in 0.1 M salt solution. In mixed membranes it can be down-shifted, for instance, to 7.9 in PC host bilayers [41]. This down-shift can be even more pronounced when the membrane contains suitable hydrogen-bonding partners, e.g., there is a shift to ≈7 in presence of PE [41]. To determine if a lipid interface of PA or PIP is indeed also more sensitive to ACh, we repeated our experiments with monolayers made from DMPA (Figure 4). Titration of the membrane indicated that $pK_{HPO_{4}^{-}}\approx 8.5$, which is in very good agreement with the literature [43]. In the regime of the pK, even more pronounced changes of lateral pressure as compared to PS took place (Δπ≥25 mN/m between pH 7.5 and 9.5). It has to be noted, however, that the “true” pK may differ slightly from the experimentally obtained one (Figure 4b). The titration curve was neither fully reversible in the case of PS nor in case of PA. i.e., back-titration with an equivalent amount of NaOH did not restore the initial state, but rather a state with lower lateral pressure. In all likelihood, this lack of reversibility was due to detachment of lipid into the subphase during titration. This may have slightly biased the π−pH diagram and thus the pK.

Due to its higher pK as compared to PS, the PA monolayer should respond to lower proton concentrations and therefore also smaller amounts of ACh. Accordingly, we reduced the concentration of ACh in the injection fluid by two orders of magnitude as compared to the experiment with PS. When ACh was applied in the micromolar range, the pH changes during catalytic hydrolysis were clearly smaller. Nevertheless, the proton pulse led to a very strong decrease of lateral pressure by more than 5 mN/m. A comparison with the titration curve (Figure 4b) indicated that, like in case of PS, this response was mainly due to headgroup protonation. When AChE was replaced by BSA, there was no step-like response of lateral pressure upon injection of ACh (Appendix A). There was also no step-like pH change. We only observed gradual acidification of the subphase. In all likelihood, this is due to a combination of spontaneous hydrolysis of ACh and the dissolution of carbon dioxide from the air. The latter leads to formation of carbonic acid which also increases the proton concentration in the system (note: the dissolution of carbon dioxide and ensuing change of pH is also evident in the experiment with AChE, where it manifests as a gradual decrease of pH from ∼300 s to 6000 s).

Taken together, these results confirmed that the lipid headgroup pK determines the range of maximal pH sensitivity. A cell membrane residing at pH∼7.5 could become very responsive to protons by incorporating lipids with pKs close to 7 such as PA and PIP. Such an interface will be sensitive to micromolar concentrations of protons as well as ACh. Intriguingly, these lipids are indeed present in postsynaptic cholinergic membranes (Table 1).

#### 3.3.2. The AChR-Lipid Interface as a Proton Receptor

The present analysis obviously raises the question: which role does the receptor protein play in the postsynaptic membrane? In the following, we suggest that AChR is involved in postsynaptic excitation by a different mechanism as compared to the presently conceived one (ligand-gated opening of a proteinous pore). There are several lines of evidence which indicate that AChR optimizes the postsynaptic membrane response to protons:(i)anionic lipids assemble around AChR,(ii)AChR does not function as a receptor for ACh in absence of anionic lipids,(iii)cell membranes containing AChR have a pK in vicinity of 7.5, and(iv)cells containing AChR-like molecules are excited by protons

These points shall be discussed in more detail: (i) Several studies have shown that anionic lipids accumulate around AChR. Evidence for this has come from spin label experiments which showed that negatively charged lipids like PA, PI, CL and to a lesser extent PS preferentially partition into the lipid annulus of AChR [44,45,46]. First, this indicates that there is a gradient of lipid species and therefore of headgroup pKs along the postsynaptic membrane. Second, if AChR has an affinity for certain anionic lipids, it will attract these molecules from the cellular lipid pool into the postsynaptic membrane.

(ii) Several independent groups have reported that AChR is “nonfunctional” when reconstituted in membranes made from nonpolar and zwitterionic lipids [47,48,49,50]. In PC:CHOL (75:25 mol%) or PC:CHOL:PE (50:25:25 mol%) AChR

“*completely lacks the ability to activate the characteristic cation-channel in response to the presence of cholinergic agonists”*.[47]

This is a striking result, because it indicates that the receptor does not function as such in a membrane interface which correctly represents ∼85–90% of the postsynaptic membrane. If protonation plays a role in membrane excitation, however, this result is expected, because PC and PE with low pKs (Table 1) as well as cholesterol with very high pK are not good proton acceptors close to pH 7.5. This finding also indicates *that the AChR molecule and its protonatable residues-by themselves-do not confer protonation sensitivity upon a membrane*. Rather, anionic lipids (PS, SM, PI, CL and PA) or a combination of them with AChR seem to be crucial for membrane excitation.

Receptor reconstitution studies also demonstrated that addition of anionic lipids to PC:CHOL mixtures can establish the ligand-gated ion-flux response [43,47,48,51]. PA seems to be particularly effective in this regard [44,45]. Membranes containing PA exhibit agonist-induced ion flux response and the latter is modulated by pH, being high at pH 8 and reduced upon acidification towards pH 6 [43]. Furthermore, it was demonstrated that the pK of PA is down-shifted from 8.5 to 7 in presence of AChR. This finding is intriguing, because it means that anionic lipids are not only accumulated around AChR but their headgroup pK may also be “tuned” by the protein. The interpretation of these reconstitution experiments, however, is not entirely straightforward, because different lipid matrices may incorporate different amounts of AChR; protein orientation in the bilayer may vary as a function of lipid and the characteristics of the vesicle (size, permeability, etc.) could influence the ion-flux response [9].

(iii) AChR covers ∼20% of the postsynaptic membrane area (Table 2) and therefore is one of the main constituents of this material. This protein has an isoelectric point of ∼5 [52] which suggests that is has one or more pKs in the vicinity of pH∼7. The existence of anionic side chains at physiological pH is also evidenced by its large calcium binding capacity [53]. Indeed, direct evidence for proton sensitivity of membranes containing AChR has come from cellular studies. When *Torpedo* or mouse AChR were expressed in *Xenopus* oocytes, extracellular application of ACh resulted in depolarizing transmembrane currents [10,54,55]. Remarkably, these currents had a strong pH dependency. Acidification led to a sigmoidal decrease of the *Torpedo* AChR current by more than an order of magnitude when going from the maximum at pH 8 to pH 6. For mouse AChR, the pH dependency was bell-shaped with a maximum at pH 7 and a steep decrease upon acidification to pH 5. These results are important, because they indicate that these membranes have a pK of approximately 7 and 6, respectively. Since the pK is a transition and since processes usually slow down in critical regimes, it would be expected that the relaxation times of the transmembrane currents also increase in these regimes. Indeed, the decay time constants of the oocyte currents were shown to be maximal at pH∼8 (*Torpedo* AChR) and ∼7 (mouse AChR) respectively. Importantly, identical findings have been reported for native cholinergic membranes. Using neuromuscular junctions, Landau et al. demonstrated that ACh-evoked miniature endplate currents (mepcs) are maximal at pH 6–7 [56]. These currents were also strongly reduced upon acidification to pH ≤5. Like in the expression studies, it was found that a maximum in mepc amplitude correlates with a maximum in decay time. *Taken together, these findings demonstrate that excitation of cholinergic membranes is strongly pH-dependent in the range 5–8, i.e., membranes containing AChR are likely to have a pK close to ∼7.*

(iv) It has been shown directly that expression of proteins from the AChR-family facilitates excitation of cells by protons [57]. When a bacterial homologue of AChR was expressed in oocytes and HEK cells, extracellular acidification resulted in robust transmembrane currents as compared to controls. A closer analysis of these currents indicated a half-maximal response at pH∼5.5.

In summary, several reports have indicated that excitation by ACh requires anionic lipids and that the latter preferentially assemble around AChR. AChR-rich membranes have a pK close to 7 and therefore should be proton sensitive at the physiological extracellular pH.

### 3.4. Acidification Pulse in the Synapse

A strong membrane response to protons requires the existence of a protonation transition. However, there also has to be a sufficiently large acidification pulse to reach these pK(s). The pH changes in a synapse depend on the quantity of ACh that is released. The concentration of ACh can be as large as ∼0.3 mM in the local vicinity of the sites of release [58]. When averaged over the entire synapse, it is probably ∼1–20 μM [58,59,60,61,62]. At these concentrations, the turnover rate of AChE is only ∼2% of its maximum [60]. Nevertheless, the released ACh is hydrolysed within <1 ms [62]. Since hydrolysis is strongly favoured, ${\left[AH\right]}_{eq}\approx {\left[ACh\right]}_{released}$. The dissociation constant of AH is $K_{a}=10^{-4.75}$. Despite being a weak acid, acetic acid is almost fully dissociated at these concentrations/pH values. This means that ∼3 × 10${}^{6}$ hydronium ions are released in the synapse (for rat diaphragm with $V_{syn}\approx$ 450 μm${}^{3}$ [62]). This also means that the pH of the synapse may drop to 5–6. Bulk as well as surface molecules, however, will readily react with the liberated protons, i.e., the synapse is strongly buffered. For the aqueous bulk this means that the pH remains widely constant. *However, for the buffering molecules it means that their charge state will change.*

**Table 2 membranes-12-00005-t002:** Approximate mole and area fractions as well as 2D-densities of phospholipids (phosphatidylserine (PS), phosphatidic acid (PA) and phosphorylated phosphatidylinositol (PIP)), cholesterol as well as AChR in postsynaptic membrane.

Material	Mole Fraction ${}^{a}$	Area Fraction ${}^{b}$	Molecules [μm${}^{-2}$]
phospholipids	0.612	0.634	1.3 $\times 10^{6}$
PS	0.086	0.089	$1.8\times 10^{5}$
PA	0.008	0.008	$1.6\times 10^{4}$
PIP	0.001	0.001	$1.3\times 10^{3}$
cholesterol	0.383	0.159	$8.0\times 10^{5}$
AChR	0.005	0.207	$2.1\times 10^{4}$
AChR ${}^{c}$		0.17	∼10${}^{4}$

${}^{a}$ based on data from Table 1 with mole fraction $\chi_{i}=\frac{n_{i}}{\sum_{z=1}^{z}n_{z}}$. ${}^{b}$ based on $\chi_{i}$ and typical area per molecule in bilayer $(a_{PL}\approx$ 0.5 nm${}^{2}$ [63], $a_{CHOL}\approx$ 0.2 nm${}^{2}$ [63] and $a_{AChR}\approx$ 10 nm${}^{2}$ [64]) with area fraction $\Gamma_{i}=\frac{A_{i}}{\sum_{z=1}^{z}A_{z}}$. ${}^{c}$ data from electron microscopic investigations [62,65].

The synaptic molecules with a pK close to the resting pH will be the most important proton acceptors. In the extracellular fluid these are probably dissolved phosphates (*c*∼0.5 mM with pK≈ 7.2 [66]) and bicarbonate (*c*∼10 mM with pK≈6.3 [67]). Although the carbonic acid-bicarbonate system is considered one of the most important biological buffers, it requires carbonic anhydrase in order to sequester protons on short time scales at pH 7 [66,67]. It is not known if this condition is fulfilled in cholinergic synapses.

Unavoidably, an acidification wave will also protonate the membrane molecules. Based on their two-dimensional densities (Table 2) and the geometry of the postsynapse ($A_{syn}\approx$ 7000 μm${}^{2}$ [62]), membrane moieties are present in the ∼1–100 mM range. The importance of these interfacial buffers as proton acceptors becomes even more apparent when comparing the ratio of bulk (e.g., dissolved phosphate) to membrane molecules (e.g., PL). In the present experiments (Figure 2, Figure 3 and Figure 4), for instance, the membrane/bulk ratio was ∼$\frac{1}{2500}$ while in a synapse it is ∼$\frac{70}{1}$. This means that in a system like the synapse with high surface-volume-ratio, membrane molecules are likely the dominant proton acceptors. Proton distribution towards the membrane interface will be aided by the high lateral density of anionic membrane moieties. The latter confer a surface potential on the interface which will additionally attract synaptic protons. The residence time of a proton (τ) on a membrane molecule will depend on the pK. For most phosphates of lipid headgroups with pK∼1–3 (Table 1) τ∼1–100 ns [42]. Therefore, most phosphates will rapidly transfer their $H^{+}$ to neighbouring molecules or to molecules in the bulk [68]. There will also be proton permeation through the bilayer which constitutes a depolarizing transmembrane current [69]. Eventually, protons will accumulate on interfacial groups with pKs >3, because this is thermodynamically favourable and because these molecules exist at high densities. Such moieties are the carboxylate of PS and the phosphate heads of PIP and PA (Table 1) as well as amino acid side chains of proteins, e.g., histidine with pK∼6.5, cystein with pK∼6.3 and the N-terminal amino group with pK∼7.6 [70].

### 3.5. Postsynaptic Excitation by Protonation-Hypothesis and Testable Predictions

Kaufmann and we have proposed previously that postsynaptic excitation is due to catalytic membrane protonation [11,14,15,16,71]. The minimal set of necessary components are ACh, esterase activity (e.g., AChE, BuChE) and a pH-responsive membrane interface. The present results (Figure 2, Figure 3 and Figure 4), experiments with lipid bilayers [12] and experiments with excitable cells [16] are in favour of this suggestion. It has not been investigated to date whether this mechanism is realized in cholinergic synapses. In order to facilitate such scrutiny, we shortly recapitulate our working model and its predictions: a proton pulse in a cholinergic synapse is inevitable and rapid due to the presence of some of the fastest catalysts that have been found in nature (AChE and BuChE). Based on the reported transmitter concentrations in the synapse [58,59,60,62], the number of synaptic protons increases from ∼10${}^{4}$ at rest to ∼3 × 10${}^{6}$ during transmission. The liberated protons will distribute in the cleft via surface and bulk pathways [42,68]. While membrane moieties with low pKs will aid in lateral proton transfer, those with a pK closer to 7.5 will transiently accumulate protons. Based on their ionization constants (Table 1) and their 2D-densities (Table 2), phospholipids (e.g., PA, phosphorylated PI, and to a lesser extent PS) as well as amino acid side chains of proteins [70] are likely to be such postsynaptic “proton sinks”. Intriguingly, several lines of evidence indicate that PA and PI are also accumulated in the annulus of AChR [44,45,46] and therefore will be concentrated in membrane domains (*c.f.* [65]). The interaction with AChR may further tune the pK of these lipid headgroups [43]. When membrane molecules are protonated, their charge state changes (Figure 2, Figure 3 and Figure 4). This can increase the probability of pore formation in the bilayer matrix and at the lipid-protein boundary. Through such defects, depolarizing membrane currents carried by protons or other ions can flow [12,15,71,72,73,74]. The ensuing change of transmembrane potential will amplify the membrane state change. The number of liberated protons sets the scale for the maximum number of membrane protonation events. This can be compared to the typical number of unitary currents which arise during postsynaptic excitation. The latter have been estimated to be ∼3 × 10${}^{5}$ [75]. This number of sites is one order of magnitude lower than the number of available protons (∼3 × 10${}^{6}$). Importantly, there also is a sufficient number of membrane molecules with a pK close to 7 that can accept these protons ($n_{PA}+n_{PIP}$ in synapse ∼10${}^{8}$; see Table 2). Therefore, proton-induced membrane excitation is quantitatively possible.

Our working model furthermore explains the time scales of the synapse. If protonation of a membrane molecule with a pK close to 7 is associated with membrane defect formation, a current may flow as long as the molecule remains protonated. For molecules with a pK close to 7, the proton dwell times are in the ∼1 ms range [42]. This indeed matches the typical time scales of excitatory currents in neuromuscular junctions (∼1–10 ms [54,56]). Aside from agreeing with the time scales, the existence of a pK close to 7 in the postsynaptic membrane would explain the pH dependency of synaptic currents. The latter are maximal at pH∼6–7 and strongly reduced upon acidification [56]. Such a reduction of current is expected if the involved molecules are taken through their protonation transition (i.e., become fully protonated and therefore non-responsive to protons under acidic conditions [15,20]). When the interfacial pK is indeed in vicinity of the resting pH, this has additional implications. Since the system is close to a transition (*c.f.* Figure 3 and Figure 4) the membrane response to protons is optimized, i.e., the system will be very pH sensitive. In addition, the fluctuation strength is generally larger [14,73,76]. This means that *the postsynapse may exhibit membrane responses in the absence of transmitter release*. The latter could be related to the phenomenon of miniature endplate potentials which at present are only attributed to spontaneous liberation of ACh into the synapse [77].

Finally, this view of the postsynapse implies that ligands can modulate the lipid-protein interaction and therefore the proton sensitivity of the membrane. Adsorption of substances (agonists, toxins, anesthetics, etc.) to AChR *and/or* the lipid membrane may change the pK as well as lateral composition of the membrane interface. α-neurotoxins, for example, bind to the rim of AChR and may thereby affect the composition of the lipid annulus, which may in turn affect proton excitation [78].

The model as proposed herein can be corroborated/falsified by relatively simple experiments. *Our hypothesis is proven wrong if acidification does not excite postsynaptic cholinergic membranes.*

## Figures and Tables

**Figure 1 membranes-12-00005-f001:**
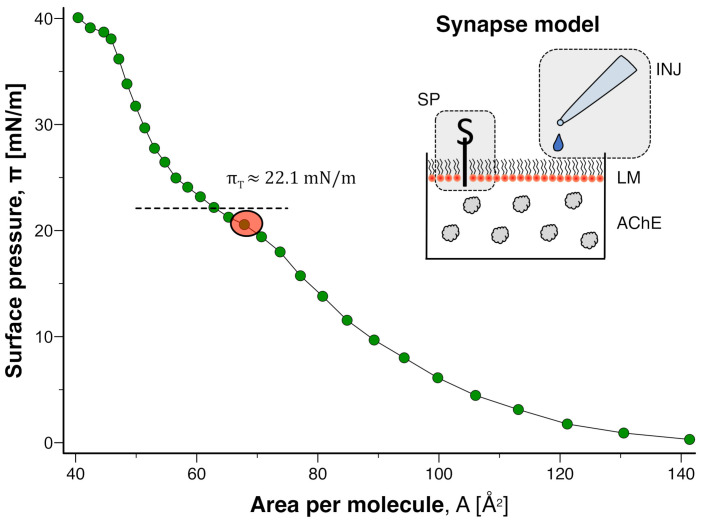
**Isothermal state diagram of DMPS monolayer on standard subphase** (100 mM NaCl, 0.5 mM phosphate buffer, pH set to 7.0 at T≈23.0 °C). *(dashed horizontal line)* Transition pressure $\pi_{T}$. *(red circle)* Initial states for injection experiments. Data series is a representative example taken from a set of n=4 measurements. *(inset)* The setup is a simple model of a cholinergic synapse. Substances can be injected *(INJ)* in vicinity of a lipid membrane *(LM)* whose state is monitored with a surface pressure sensor *(SP)*. The subphase contains acetylcholinesterase *(AChE)*.

**Figure 2 membranes-12-00005-f002:**
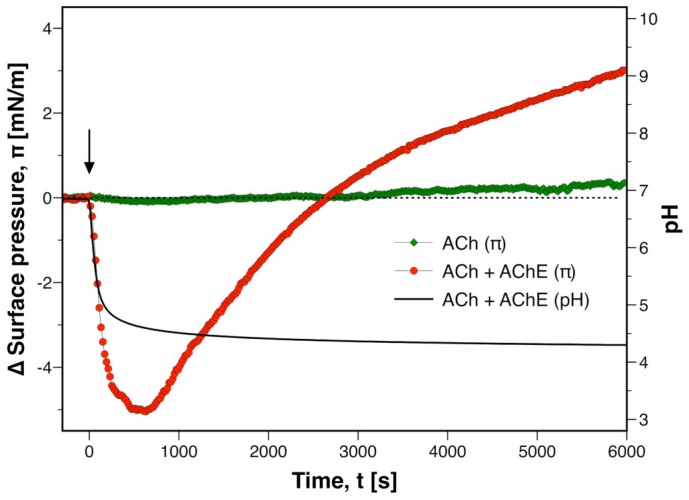
**Catalytic hydrolysis of acetylcholine (ACh) triggers state change in DMPS monolayer.** Changes of surface pressure π
*(left axis)* and subphase pH *(right axis)* when ACh was injected *(arrow)* onto a DMPS monolayer. The subphase either did *(red circles)* or did not *(green diamonds)* contain acetylcholinesterase. Each data series is a representative example taken from a set of n=5 measurements.

**Figure 3 membranes-12-00005-f003:**
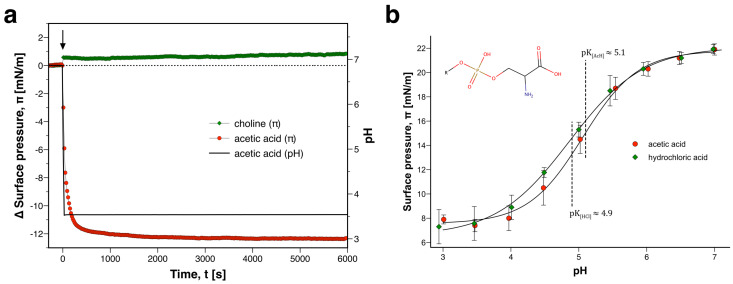
**Control experiments with hydrolysis products of ACh.** (**a**) Surface pressure change upon injection of choline *(green diamonds)* or acetic acid *(red circles)* onto a DMPS monolayer. Each data series is a representative example taken from a set of n=5 measurements. (**b**) Three protons can dissociate from the headgroup of phosphatidylserine (PS). Titrations with acetic acid *(red circles)* or hydrochloric acid *(green diamonds)* reveal the strong membrane state changes at $pK_{COOH}$. Solid lines represent fits of Boltzmann sigmoid functions. Each data series consists of mean values ± StDev of n= 3–4 measurements.

**Figure 4 membranes-12-00005-f004:**
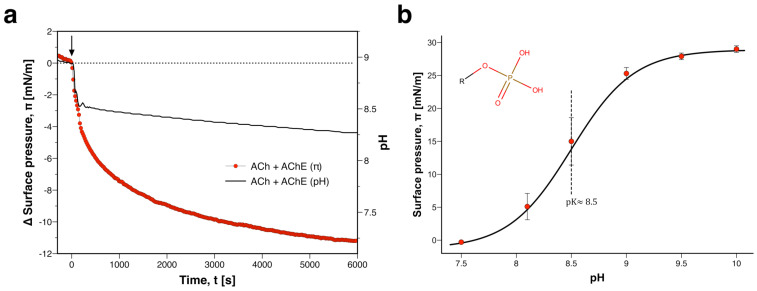
**State change in phosphatidic acid (PA) monolayer upon injection of acetylcholine (ACh).** (**a**) Change in surface pressure π
*(red circles, left axis)* and subphase pH *(line, right axis)* when ACh was injected *(arrow)* onto a DMPA monolayer with acetylcholinesterase (AChE). (**b**) Two protons can dissociate from the phosphate head of PA. Titration with hydrochloric acid reveals strong membrane state changes around $pK_{HPO_{4}^{-}}$. The solid line represents the fit of the Boltzmann sigmoid function to average data *±* StDev of n=3 measurements.

**Table 1 membranes-12-00005-t001:** Material composition of AChR-rich membranes from *Torpedo californica* and *T. marmorata* as well as apparent ionization constants of phospholipids (PL) as determined with protein-free lipid membranes in salt solution.

Material	Content ${}^{a}$	Ionization Constant(s) ${}^{b}$		
	[mol%]	$\left({pK}_{app}\right)$		
phosphatidylcholine (PC)	38–46	∼**1** $\left(HPO_{4}\right)$		
ethanolamine phosphoglyceride (EPG)	31–43	∼**3** $\left(HPO_{4}\right)$		**9.6–10.1** $\left(NH_{3}^{+}\right)$
phosphatidylserine (PS)	11–16	**0.5–2.6** $\left(HPO_{4}\right)$	**3.9–5.4** (COOH)	**9.8–11.6** $\left(NH_{3}^{+}\right)$
sphingomyelin (SM) and lyso-PC	1.5–7	$n.a.\;(HPO_{4})$		
phosphatidylinositol (PI)	*n.d.*–4	$2.5\;(HPO_{4})$	**6.5–7.7** ${\left(HPO_{4}^{-}\right)}^{In}$	
cardiolipin (CL)	*n.d.*–3	**2–2.5** ${\left(HPO_{4}\right)}^{{}^{\prime }}$	**2.5–3** ${\left(HPO_{4}\right)}^{{}^{\prime \prime }}$	
phosphatidic acid (PA)	*n.d.*–2.5	**3–4** $\left(H_{2}PO_{4}\right)$	**6.5–8.7** $\left(HPO_{4}^{-}\right)$	
PL/cholesterol [mol/mol] ${}^{c}$	≈1.6			
PL/protein [mol/mol] ${}^{d}$	≈130			

^a^ minimal to maximal values based on data from [8,29,30,31] as rounded to nearest 0.5%; *n.d.* (not determined in some of the studies). ^b^ data are from measurements in 100 mM NaCl unless noted otherwise; [32] for PC (note: no salt), [28,33] for PE in a host bilayer (PC), [20,26,27,34] and present work for PS, [28] for PS in a host bilayer (PC), [35,36,37] for phosphate on glycerol and inositol (*In*) of PI, [38,39] for the first (′) and second (″) phosphate of CL (note: 5 mM NaCl in [38], [32,34,40] for PA (note: 500 mM NaCl in [32], *Torpedo* Ringer buffer in [40]), and [40,41] for PA in host bilayer (PE), *n.a.* (data not available). ^c^ average of data from [8,29,30,31]. ^d^ average of data from [8,29,30], assuming that a typical phospholipid has a molecular weight of 750 g/mol and that the only protein present is AchR with a molecular weight of 290,000 g/mol.

## Data Availability

The data presented in this study are available in the article and in the Appendix A.

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
