# Peer review of "Lipid Membrane State Change by Catalytic Protonation and the Implications for Synaptic Transmission"

_membranes, 2021, doi:10.3390/membranes12010005_

Round 1

Reviewer 1 Report

In this study, authors investigated the response of membrane lipids phosphatidylserine (PS) and phosphatidic acid (PA) to the hydrolysis of the neurotransmitter acetylcholine (ACh). They find lateral pressure changes as a consequence of acetylcholinesterase (AChE) activity, which they link to the protonation of lipid headgroups. They propose that cholinergic transmission is due to postsynaptic membrane protonation. Although interesting, the model needs further validation as conclusions are not fully supported by the data and controls. It should be excluded that the observed effect in lateral pressure (figure 2) is specific of AchE. A control of injection of another unrelated protein with no catalytic activity should be performed. Equivalent experiments should be also done with PA. In general, control experiments with lipids that do not undergo any ionization change should be included. The effect of lipid protonation on membrane excitation and receptor (AChR) activity should be tested. In addition, I have some minor comments. Last paragraph of the intro is written as project aims in future tense. It should be rewritten in past tense. Results and discussion sessions should be separated. It is misleading to mix previous results with new results from this study.

Round 2

Reviewer 1 Report

The manuscript has improved after revision. I think the author addressed all my concerns.